# Thermo-Elasto-Hydrodynamic Characteristics Analysis of Journal Microbearing Lubricated with Rarefied Gas

**DOI:** 10.3390/mi11110955

**Published:** 2020-10-22

**Authors:** Yao Wu, Lihua Yang, Tengfei Xu, Wei Wu

**Affiliations:** 1State Key Laboratory for Strength and Vibration of Mechanical Structures, Xi’an Jiaotong University, Xi’an 710049, China; nealjackman@stu.xjtu.edu.cn (Y.W.); xtf1992@stu.xjtu.edu.cn (T.X.); wwkyou@stu.xjtu.edu.cn (W.W.); 2Shaanxi Key Laboratory of Environment and Control for Flight Vehicle, Xi’an 710049, China; 3School of Aerospace Engineering, Xi’an Jiaotong University, Xi’an 710049, China

**Keywords:** gaseous rarefaction effects, thermo-elasto-aerodynamic lubrication, viscosity–temperature effect, compliance matrix, bearing characteristics

## Abstract

Temperature rise and elastic deformation are unavoidable issues occurring in high-speed gas microbearings due to the dominant small-scale fluid dynamics in rarefied gas flow applications. In this paper, thermo-elasto-aerodynamic analysis requires simultaneously solving the modified Reynolds equation, modified energy equation, temperature–viscosity relationship and the elasticity equations for predicting the lubrication characteristics of microbearings. A thermo-elasto-aerodynamic lubrication is systematically investigated by using the partial derivative method, finite difference formulation and the finite element approach. The results indicate that, compared with rigid microbearing which has a constant viscosity gas lubricant, the temperature effect increases the load capacity, friction coefficient and stiffness coefficients, and it decreases the attitude angle and damping coefficients of the microbearing. The flexibility of the bearing pad also leads to the increase in load capacity and direct stiffness coefficients, while it remains to further decrease the direct damping coefficients on the basis of thermo-aerodynamic performance. The present study is conducive to accurately analyze the microscopic flow properties in a microbearing-rotor system.

## 1. Introduction

Along with the rapid development of microfabrication technologies, high-speed gas-lubricated microbearings are one of the vital components in various rotating microdevices and power micro-electro-mechanical system (MEMS) applications such as microturbines, microgenerators, microdrills, micropumps and magnetic storage products. They can offer certain advantages over conventional oil film bearings, rolling element bearings and active magnetic bearings in a microsystem [1,2,3]. The pressure is formed within ultralow clearance in the aerodynamic journal bearing for supporting micro rotors, as the rotation speed of the rotor comes close to hundreds of thousands of revolutions per minute, and the journal bearing gap is only ten or so microns. The application of ultra-thin gas lubrication becomes particularly valuable to increase the stability of micro-rotating machinery due to its better and feasible features. It is also known that heat generation, torsional drag and temperature distribution inside the gas journal microbearing are proportional to the lubricant viscosity. Since air viscosity is much smaller than that of oil and it does not require an external air feeder, a gas bearing that has a relatively simple structure, by comparison, would be virtually wear-free and frictionless, and it would have less parasitic power loss, no lubricant contamination, low noise and long maintenance-free life [4]. The upgrading of ultra-thin film lubrication technology also makes it possible for micro rotating machinery such as micro gas turbines, micro turbine generators, ultra-high speed blowers/compressors, intelligent micro/nano robots and micro unmanned aerial vehicles to realize various energy conversions, micro power supplies and improved energy utilization efficiency [5,6]. In conventional gas bearing analysis and design procedures, the bearing characteristics are determined by assuming that the fluid flow in the gas films is in isothermal continuum and that the bearing pads are rigid bodies. As the gas film thickness becomes thinner, the flow behaviors of the compressible gas at the micro-scale differ from those at macro-scale, and the continuum theory may lose its usefulness [7,8,9,10]. The effect of gas rarefaction is the most important factor that considerably affects the bearing performance in microfluidic devices. The level of rarefaction can be characterized by Knudsen number *K_n_*, defined as the ratio of the mean free path of gas molecules λ_0_ to the characteristic length scale *L* [11,12]. For larger Knudsen numbers, the Boltzmann equation formulation should be employed to describe the rarefied gas flow in the whole rarefaction regimes. Moreover, it is well known that at high speed, the gas film temperature over the bearing does not remain constant, and the temperature rise induces distinct variations in gas viscosity. No engineering surface is perfectly rigid, so the elastic deformation of a journal microbearing alters the gas film thickness profile to some extent, which makes the aerodynamic pressures different from those of rigid bearings. In order to guarantee the safe and reliable operating conditions of microfluidic machines, it is desirable to reveal the interactions between gas compressibility, rarefaction, temperature rise and the bearing flexibility on rarefied gas flow.

Thermo-hydrodynamic (THD) lubrication problems have been substantially studied over the past several decades. Early theoretical solutions were published by Hughes and Osterle [13], who treated the lubricant viscosity as a function of pressure and temperature in journal bearing. Similarly, McCallion et al. [14] ignored the pressure terms in the energy equation to uncouple the Reynolds equation and energy equation, and they compared the load parameter with those of the isothermal ones. The importance of bearing geometry, oil type and inlet temperature for the THD performance of slider bearings was demonstrated by Ezzat et al. [15,16]. They found a hysteresis phenomenon in the pressure–temperature relationship. Paranjpe and Han [17] proposed a coupled approach to solve the energy equation with consideration of the cavitation effects. The thermal effects of a magnetoresistive transducer were examined by Bogy et al. [18] using the heat transfer model with discontinuous boundary conditions. Khonsari et al. [19,20] developed simplified THD design charts for evaluating the stiffness and damping coefficients and threshold speed of journal bearings, and they gave the relationship between the inlet viscosity and the oil whirl stability. Stevanović and Milićev [21] studied the non-isothermal subsonic slip gas flow in slider microbearings and solved the nonlinear second-order differential system equations by assuming that the pressure, velocity and temperature are a perturbation series in terms of Knudsen number. Wang et al. [22,23] coupled the molecular gas film lubrication (MGL) equation, modified energy equation and the rotor kinetic equations to elucidate the nonlinear dynamic properties of a micro-bearing-rotor system. Zhang et al. [24,25] explored the influences of rarefaction, large temperature gradient and thermal creep on the steady characteristics of micro gas bearings for different gaseous lubricant species. All aforementioned contributions are based upon the assumption that both journal and bearing surfaces are rigid, and no elastic deformation occurs. It has also been reported in the literature that the magnitude of elastic deformation is comparable with the film thickness due to the flexibility of the bearing pad. Rohde and Oh [26] analyzed the effect of temperature on the elastic and thermal deformations of the solid as well as the fluid properties for inclined-plane slider bearing by coupling lubricant film momentum, continuity, energy equations and heat conduction equations. Khonsari and Wang [27] combined the finite difference scheme and the finite element method to compute the bearing thermoelastic deformation and journal thermal dilation in a thermo-elasto-hydrodynamic (TEHD) model.

In recent years, to obtain accurate thermal-elastic-tribological analysis, the incorporation of dynamic loading [28], turbulent flow [29], shaft and bearing thermal expansions [30], polytetrafluoroethylene (PTFE) pad liners [31,32], gas spiral groove face seals with choked flow conditions [33] and shaft misalignment effects [34] into the TEHD lubrication theory has stimulated further interest. Numerical computation and simulation are more effective in predicting detailed information about the lubrication characteristics of the bearing. For example, Yang and Palazzolo [35,36] utilized computational fluid dynamics and fluid–structure interaction (FSI) to investigate the TEHD lubrication of tilting pad journal bearings and log decrement in the Jeffcott rotor model. They concluded that the CFD model has a higher reliability compared to the Reynolds model mixing coefficient approach. Although a number of studies about the gaseous microflows problem of microbearings and hard disk drives (HDDs) have been undertaken, these works mainly focus on static lubricant film characteristics under isothermal rigid conditions inside the ultra-thin spacing. The theoretical results relating to dynamic characteristics, which are necessary for the stability of a rotor-bearing system, have received relatively little attention, most likely because of the complex non-linear property. Furthermore, the task is more complicated when the non-isothermal temperature field, the viscosity–temperature relationship and the elastic deformation of the bearing pad collectively affect the static and dynamic characteristics of actual micro gas journal bearings. Hence, a fundamental study on the thermo-elasto-hydrodynamic performances of microbearings provides guidelines for their design and operation. To better understand the flow properties and temperature rise of gas films in close spacings, it is important to explain the temperature-related phenomena in microrotating machinery.

In the condition of considering the temperature–viscosity effect of rarefied gas and the flexibility of bearing pads, this paper investigates the static and dynamic characteristics of gas-lubricated journal microbearings, including the temperature distribution of gas film, load capacity, attitude angle, friction coefficient and dynamic coefficients of bearing for various parameters. Numerical solutions of the thermo-elasto-aerodynamic performance are compared with those of the continuum flow, isothermal gas rarefaction as well as thermo-aerodynamic results in detail. The analysis is of practical importance in bearing design to improve the stability of microbearing rotor systems, which is highly desirable for microrotating machinery.

## 2. Methodology

The schematic and geometry of a gas-lubricated journal microbearing are shown in Figure 1. The fluid inertia effects are neglected, and the gas flow is assumed to be laminar, compressible and Newtonian. Generally, the shaft has a reasonably higher Young’s modulus than that of the bearing pad, and only the elastic deformation of the bearing surface under the aerodynamic air film is considered in the analysis. The generalized lubrication equation includes the gas rarefaction effects, bearing pad elasticity and temperature–viscosity effect, which is applicable for arbitrary Knudsen numbers, and can be written in nondimensional form as follows:(1)$$\frac{\partial }{\partial \varphi }\left(QPH^{3}\frac{\partial P}{\partial \varphi }\right)+\frac{\partial }{\partial \eta }\left(QPH^{3}\frac{\partial P}{\partial \eta }\right)=\Lambda_{b}\frac{\partial (PH)}{\partial \varphi }+2\Lambda_{b}\frac{\partial (PH)}{\partial \tau }$$
where *P* = *p*/*p_a_* and *H* = *h*/*c* are non-dimensional pressure and film thickness between the bearing and journal. *φ* = *x*/*R* and *η* = *z*/*R* are the circumferential and axial coordinates, and *x* and *y* are local Cartesian coordinates in the length and width directions. *p_a_* is the ambient atmospheric pressure, *c* is the radial clearance, *R* is the journal radius and *p* and *h* = *c* + *e*cos*φ* are the dimensional pressure and film thickness of the gas film. *e* is the eccentricity, *ε* is the eccentricity ratio and *ε = e*/*c*. Λ*_b_* = 6*μ*(*T*)*ωR*^2^/(*p_a_c*^2^) is the bearing number. The gas lubricant viscosity *μ*(*T*) varies as a function of temperature, and *T* is the gas film temperature. *ω* is the angular velocity of the shaft, and *τ* is dimensionless time. *Q* is the Poiseuille flow rate ratio.

As shown in Figure 2, based on the reference Hwang et al. [37], according to the kinetic theory, the high-order pressure-driven flow (Poiseuille flow) velocity distributions *u* and *w* between two parallel plates, separated by distance *h*, are deduced as
(2)$$\left\{\begin{array}{l} u=U\left(1-\frac{y+a\lambda_{0}}{h+2a\lambda_{0}}\right)-\frac{1}{2\mu }\frac{\partial p}{\partial x}\left(b_{1}D^{c_{1}}h^{2}+a_{1}\lambda_{0}h+hy-y^{2}\right)\\ w=-\frac{1}{2\mu }\frac{\partial p}{\partial z}\left(b_{1}D^{c_{1}}h^{2}+a_{1}\lambda_{0}h+hy-y^{2}\right) \end{array}\right.$$
where *a* is the surface correction coefficient, which usually is taken to be 1. *U* is the sliding velocity of the bottom moving surface. *a*_1_, *b*_1_ and *c*_1_ are the three adjustable coefficients.

The normalized Poiseuille flow rate *Q*_P_ in ultra-thin gas film and the flow rate coefficient of continuum flow *Q*_con_ are expressed as
(3)$$Q_{P}=\frac{\frac{h^{3}}{2\mu }\frac{\partial p}{\partial x}\left(b_{1}D^{c_{1}}+\frac{a_{1}\sqrt{\pi }}{2D}+\frac{1}{6}\right)}{-\frac{h^{3}}{2\mu D}\cdot \frac{\partial p}{\partial x}}=\frac{D}{6}+\frac{a_{1}\sqrt{\pi }}{2}+b_{1}\cdot D^{c_{1}+1},Q_{\mathrm{con}}=\frac{D}{6}$$
where $D=\frac{\sqrt{\pi }}{2K_{n}}$ is the inverse Knudsen number, and *K_n_* is the Knudsen number.

For *a*_1_ = 0.01807, *b*_1_ = 1.35355 and *c*_1_ = −1.17468, the Poiseuille flow rate ratio *Q* [38] is simplified to
(4)$$Q=\frac{Q_{P}}{Q_{\mathrm{con}}}=1+0.10842K_{n}+9.3593/K_{n}{}^{-1.17468}$$

The transient term *∂*(*PH*)/*∂τ* can be eliminated at steady-state conditions, and the modified Reynolds equation for the static characteristics of a gas-lubricated journal microbearing is then given by
(5)$$\frac{\partial }{\partial \varphi }\left(QPH^{3}\frac{\partial P}{\partial \varphi }\right)+\frac{\partial }{\partial \eta }\left(QPH^{3}\frac{\partial P}{\partial \eta }\right)=\Lambda_{b}\frac{\partial (PH)}{\partial \varphi }$$

Equation (5) can be solved by the finite element method and the successive over relaxation method. The details of the main solution process used in this paper can be found in reference [39].

The dimensionless gas film forces generated in the microbearing are obtained by integrating the film pressure along both horizontal and vertical directions.
(6)$$\left\{\begin{array}{l} {\bar{F}}_{x}=p_{a}R^{2}\int_{-\frac{B}{2R}}^{\frac{B}{2R}}\int_{0}^{2\pi }(P-1)\sin\varphi d\varphi d\lambda \\ {\bar{F}}_{y}=p_{a}R^{2}\int_{-\frac{B}{2R}}^{\frac{B}{2R}}\int_{0}^{2\pi }(P-1)\cos\varphi d\varphi d\lambda \end{array}\right.$$
where *B* is the bearing width.

The attitude angle *θ* is calculated by
(7)$$\theta =\arctan\left(\frac{{\bar{F}}_{x}}{{\bar{F}}_{y}}\right)$$
*W* is the load capacity of the microbearing, and $W=\sqrt{{\bar{F}}_{x}{}^{2}+{\bar{F}}_{y}{}^{2}}$. The non-dimensional load-carrying capacity is written as
(8)$$C_{L}=\frac{W}{p_{a}RB}=\frac{R}{B}\int_{-\frac{B}{2R}}^{\frac{B}{2R}}\int_{0}^{2\pi }(P-1)\cos\varphi d\varphi d\lambda$$

The friction coefficient on the journal surface can be computed by
(9)$$F_{b}=-\int_{-\frac{B}{2R}}^{\frac{B}{2R}}\int_{0}^{2\pi }\left(\frac{\Lambda }{6}\frac{1}{H}+\frac{H}{2}\frac{\partial P}{\partial \varphi }\right)d\varphi d\lambda$$

After the static pressure and gas film thickness are obtained, according to the small perturbation method based on the linearization hypothesis, the time-dependent modified Reynolds Equation (1) for ultra-thin gas lubrication of a journal microbearing is solved by the partial derivative method [40,41], and the dynamic coefficients *K_ij_* and *D_ij_* (*i*, *j* = *x*, *y*) that depend on the excitation frequency are computed. The detailed solution process is given in reference [42].

## 3. Modified Energy Equation and Microbearing Flexibility

In the thermo-elasto-aerodynamic analysis, gas film pressure, lubricant viscosity and elastic deformation of the bearing pad as well as gas film temperature are interdependent variables. The combined effects of these influence factors should be considered in the entire iterative numerical process. In order to obtain the temperature field through the gas film, an energy equation must be formulated and solved. Considering the small gap between the rotating shaft and the microbearing, the energy equation needs to be modified for various gas rarefaction regions.

The work and heat in gas flow are produced by the flow resistance, the shear stress on the shaft surface and the heat transfer at the journal–lubricant–bearing interface. It was found that only a little part of heat is evacuated by conduction in the solids, and approximately 90% of heat is carried away by the fluid film in most practical problems [17,43,44,45]. The length-to-diameter ratio of the gas microbearing is one order of magnitude lower than that of a typical gas journal bearing. In this paper, we assume that the lubricant flow is adiabatic, and there is no heat transfer from the rarefied gas to the journal and bearing pad. As shown in Figure 2, since the spacing in the bearing thickness direction is much smaller than the length scales in the horizontal directions, the conduction term in the *y* direction ∂*T*/∂*y* is ignored. The modified energy equation including the effects of rarefaction, compressibility and microbearing pad flexibility can be expressed as
(10)$${\bar{q}}_{x}\frac{\partial T}{\partial x}+{\bar{q}}_{z}\frac{\partial T}{\partial z}=\frac{\frac{U^{2}\mu (T)}{h+2\lambda_{0}}+\frac{h^{3}}{2\mu (T)}\frac{Q_{P}}{D}\left[{\left(\frac{\partial p}{\partial x}\right)}^{2}+{\left(\frac{\partial p}{\partial z}\right)}^{2}\right]}{J\rho c_{v}}$$
where *c_v_* is the specific heat of the lubricant, and *ρ* is the fluid density. *J* is the mechanical equivalent of heat, and *T* is the gas film temperature. ${\bar{q}}_{x}$ and ${\bar{q}}_{z}$ are the volume flow rate in the sliding and width directions, respectively.
$$\left\{\begin{array}{l} {\bar{q}}_{x}=\int_{0}^{h}udy=\frac{Uh}{2}-\frac{h^{3}}{2\mu }\frac{\partial p}{\partial x}\left(b_{1}D^{c_{1}}+\frac{a_{1}\sqrt{\pi }}{2D}+\frac{1}{6}\right)\\ {\bar{q}}_{z}=\int_{0}^{h}wdy=-\frac{h^{3}}{2\mu }\frac{\partial p}{\partial z}(b_{1}D^{c_{1}}+\frac{a_{1}\sqrt{\pi }}{2D}+\frac{1}{6}) \end{array}\right.$$

Introducing the following nondimensional variables
(11)$$\mu =\mu_{0}\mu^{*},\;{\bar{q}}_{x}=UcQ_{x},\;{\bar{q}}_{z}=UcQ_{z},\;T=\frac{UR\mu_{0}}{2J\rho c_{v}c^{2}}T^{*},\;U=\omega R$$
where *μ** is the dimensionless gas viscosity, *Q_x_* and *Q_z_* are the normalized flow rate in the circumferential and axial direction, *T** represents the dimensionless temperature and *μ*_0_ is the gas dynamic viscosity at temperature *T*_0_.

Thus, the modified energy equation becomes
(12)$$Q_{x}\frac{\partial T^{*}}{\partial \varphi }+Q_{z}\frac{\partial T^{*}}{\partial \eta }=\frac{2\mu^{*}}{H+2K_{n}{}^{*}}+\frac{6H^{3}}{\Lambda_{b}{}^{2}\mu^{*}}Q\left[{\left(\frac{\partial P}{\partial \varphi }\right)}^{2}+{\left(\frac{\partial P}{\partial \eta }\right)}^{2}\right]$$
where
$$\left\{\begin{array}{l} Q_{x}=\frac{H}{2}-\frac{H^{3}}{2\Lambda_{b}\mu^{*}}\frac{\partial P}{\partial \varphi }Q\\ Q_{z}=-\frac{H^{3}}{2\Lambda_{b}\mu^{*}}\frac{\partial P}{\partial \eta }Q \end{array}\right..$$

An iterative finite difference method is applied in the discretization of the modified energy equation for obtaining the temperature distribution in the circumferential and axial directions. The microbearing domain is discretized into a set of nodes, and point (*i*, *j*) stands for a discretization point. Equation (12) is discretized by the central difference scheme to ensure the computational accuracy and is solved by the Newton–Raphson algorithm.

Therefore, the difference approximation of dimensionless modified energy equation can be written as
(13)$$\begin{array}{l} \left(\frac{Q_{x(i,j)}}{\varphi_{i}-\varphi_{i-1}}+\frac{Q_{z(i,j)}}{\eta_{j}-\eta_{j-1}}\right){T^{*}}_{i,j}=\frac{Q_{x(i,j)}}{\varphi_{i}-\varphi_{i-1}}{T^{*}}_{i-1,j}+\frac{Q_{z}{}_{\left(i,j\right)}}{\eta_{j}-\eta_{j-1}}{T^{*}}_{i,j-1}+\\ \frac{2{\mu^{*}}_{i,j}}{H_{i,j}+2K_{n}{{}^{*}}_{\left(i,j\right)}}+\frac{6H_{i,j}{}^{3}}{\Lambda_{b}{{}_{i,j}}^{2}\mu^{*}{}_{i,j}}Q_{i,j}\left[{\left(\frac{P_{i,j}-P_{i-1.j}}{\varphi_{i}-\varphi_{i-1}}\right)}^{2}+{\left(\frac{P_{i,j}-P_{i,j-1}}{\eta_{j}-\eta_{j-1}}\right)}^{2}\right] \end{array}$$
where
$$\left\{\begin{array}{l} Q_{x}{}_{\left(i,j\right)}=\frac{H_{i,j}}{2}-\frac{H_{i,j}{}^{3}}{2\Lambda_{b}{}_{i,j}\mu^{*}{}_{i,j}}Q_{i,j}\frac{P_{i,j}-P_{i-1.j}}{\varphi_{i}-\varphi_{i-1}}\\ Q_{z}{}_{\left(i,j\right)}=-\frac{H_{i,j}{}^{3}}{2\Lambda_{b}{}_{i,j}\mu^{*}{}_{i,j}}Q_{i,j}\frac{P_{i,j}-P_{i,j-1}}{\eta_{j}-\eta_{j-1}} \end{array}\right.,Q_{i,j}=1+0.10842K_{n}{}_{\left(i,j\right)}+9.3593/K_{n}{{}_{\left(i,j\right)}}^{-1.17468}$$

The following temperature boundary conditions are used for the energy equation solution:
(14)$$\left\{\begin{array}{l} T\left|{}_{\varphi =0,\eta }\right.=T\left|{}_{\varphi =2\pi ,\eta }\right.,\\ \frac{\partial T}{\partial \eta }\left|{}_{\eta =0}\right.=0 \end{array}\right.$$

The temperature rise of the lubricant induces the important cross-film viscosity variation. It is assumed that variation of viscosity due to pressure is negligible compared to that due to temperature increase; thus, gas viscosity is only a function of temperature. The viscosity–temperature relationship is given by the well-known Sutherland law formulation:
(15)$$\mu (T)=\mu_{0}{\left(\frac{T}{T_{0}}\right)}^{1.5}\left(\frac{T_{0}+T_{s}}{T+T_{s}}\right)$$
where *T*_s_ is the Sutherland constant that is related to the gas species.

Elasticity equations in a solid domain are solved in the elastic deformation analysis of the bearing pad. The radial component of elastic deformation at the lubricant–bearing interface is needed for the modification of gas film thickness. The elastic deformation of the microbearing surface with elastic modulus *E* and Poisson ratio *υ* under aerodynamic load is calculated through a realistic compliance matrix, which is established by the finite element approach. The bearing pad is divided by eight-noded hexahedral linear iso-parametric elements, and the product of rows and columns of the compliance matrix is equal to the number of all nodes on the microbearing surface. The radial elastic deformation *δ_t_* on the working surface of the bearing pad in response to the aerodynamic pressure can be written as
(16)$$\delta_{t}=\sum C\cdot P,\;C=K^{-1}$$
where *C* is the compliance matrix, and *P* is the gas pressure matrix. *K* is the global stiffness matrix of the microbearing.

For a flexible bearing experiencing elastic distortion, the modified film thickness *H* at any angle *θ* is the sum of the rigid film profile and elastic deformation of the bearing pad.
(17)$$H=H_{0}+\delta_{t}=1+\varepsilon \cos(\varphi -\theta )+\delta_{t}$$
where *H*_0_ is the static gas film thickness in the thin film flow region, and *δ_t_* is the elastic deformation of the bearing pad surface under gas film pressure.

The boundary conditions for elasticity equations are as follows:(1)The microbearing pad is considered to be a three-dimensional cylindrical structure of finite length enclosed in a rigid housing, and the nodal displacement components on the outer surface of bearing pad that are in contact with the housing are taken as zero.(2)The bearing pad is subjected to the distributed load *P*, namely the pressure load is acting on all the nodes of microbearing working surface according to a given sequence of nodes on the bearing surface.(3)The radial displacements of the pad–housing interface are continuous and periodic in the circumferential direction because the starting and ending planes in the finite element model of the microbearing are identical.

Computer programming and simulation processes are usually challenging in thermo-elasto-aerodynamic lubrication. Firstly, the modified Reynolds equation and the elasticity equations are solved at the same time with the initial value of the viscosity and temperature until the gas film pressure and the solid elastic deformation are determined. With distributions of gas film thickness and pressure updated, the modified energy equation and the changes of lubricant viscosity at each node with temperature are computed. Once the temperature and viscosity field are evaluated, then the program turns back to the modified Reynolds equation. They are solved iteratively until convergence.

It is well-known that the rarefaction effect, elastic deformation of the bearing surface and thermal effects in a fluid lubricating film can significantly affect the critical speed, unbalance response and nonlinear dynamic characteristics of an aerodynamic microbearing-rotor system in practice. The comprehensive thermo-elasto-hydrodynamic analysis of gas journal microbearings is crucial to obtain accurate performance predictions of microfluidic or vacuum devices in high-speed micro-turbomachinery, the main findings presented in the paper will further deepen the understanding of journal–bearing interaction mechanisms for the rarefied gas lubrication problems.

## 4. Results and Discussion

The mathematical model mentioned above is used to analyze the performances of a gas-lubricated journal microbearing system. The temperature distribution inside the lubricant, load carrying capacity, attitude angle, friction coefficient, direct stiffness and damping coefficients are given for different journal sliding speeds, eccentricity ratios and elastic modulus *E*, and so forth. Table 1 lists the detailed parameters of the gas journal microbearing in this research work.

To verify the present model and the corresponding solution algorithm, the circumferential pressure distributions of the lubricant film with and without slip flow are compared with the published results, as shown in Figure 3. It is noted that the numerical results of this paper are in close agreement with the theoretical analysis by Orr [46] and Lee et al. [47] for *ε* = 0.8, *B/D* = 0.075 and *Λ* = 1, which show the validity of the calculation procedure.

### 4.1. Steady-State Characteristics

The changes to the three-dimensional temperature distributions and the contours of the temperature field with journal sliding speed *ω*, eccentricity ratio *ε* and the compliant bearing pad are shown in Figure 4. The contour plots more clearly show the details of temperature variations over the bearing areas. It reveals that microbearings presented a similar temperature distribution as shaft speed and eccentricity ratio increased. It has been assumed that all heat generated by the ultra-thin lubricating film has been carried away in the rarefied gas, the journal and microbearing pad being perfect insulators, and the lubricant flow is referred to as adiabatic flow. The film pressure first increased to the maximum value and then decreased with the circumferential coordinate *φ*, and the peak cross-film temperature emerged around the location of the minimum film thickness for aerodynamic lubrication. These results are attributed to the decrease of steady film thickness and the accentuated aerodynamic effect. Since a thinner gas film is associated with higher viscous dissipation, a stronger pressure gradient will develop across the film. The larger eccentricity ratio yielded a relatively greater temperature increase than that at higher rotational speed. It is also observed that there was a slight decrease in the temperature profile *T* as the pad became more flexible. This behavior can be explained by surface deformation of the microbearing pad, which creates more space for the gas lubricant at the loaded bearing zone.

Figure 5 and Figure 6 present the variations of the load capacity *C_L_* and the attitude angle *θ* with eccentricity ratio *ε* for continuum flow, gas rarefaction and thermo-aerodynamic and thermo-elasto-aerodynamic models, respectively. In all cases, it can be seen that the load capacities increased with the increase of eccentricity ratio, while the attitude angles decreased. The load capacity of the continuum model was larger than those of other models. When the temperature effect was considered, the increase of the gas temperature led to a larger gas viscosity, and then the load-carrying capacity in the thermo-aerodynamic and thermo-elasto-aerodynamic cases were higher than those when only considering the impact of gaseous rarefaction effects. As the elastic modulus of the microbearing pad increased from 5 to 200 GPa, the dimensionless load capacities increased marginally at the small eccentricity ratio. On the contrary, *C_L_* increased gradually as *E* decreased for *ε* > 0.7. This is because the surface deformation of the microbearing increases the thin film gap, which dilutes the influence of rarefaction. The corresponding changes to pressure, temperature and film thickness are depicted in Figure 7 and Figure 8. The increase of gas film thickness due to elastic deformation of the bearing pad led to a decreased gas film pressure and temperature rise at *ε* = 0.3. A small kink in the film thickness profile of microbearing at *E* = 5 GPa and *ε* = 0.7 caused the reverse tendency in the pressure contour so that it was similar to those encountered in load capacity. The effects of temperature and elastic distortion on the attitude angle are obvious. The attitude angle in the thermo-aerodynamic case was larger than that in cases of thermo-elasto-aerodynamic lubrication, and the attitude angle decreased significantly by increasing the bearing pad flexibility at a higher eccentricity ratio range. With the increase of the eccentricity ratio, the load capacity became larger while the attitude angle was smaller in the gas film.

The variation of the load carrying capacity with the rotating speed of the rotor is described in Figure 9 for both conditions of with and without the elastic deformation and thermal effects considered. It is found that the load capacity was near linearly proportional to the rotation speed. In comparison with the gas rarefaction case, the temperature and elastic deformation effects increased the magnitude of *C_L_*, especially for higher values of the rotation speed, which indicates the compressibility effect and the aerodynamic phenomenon are more remarkable. Increasing the rotation speed enhanced the friction coefficient of the journal surface, as depicted in Figure 10. At the same *ω*, the value of the friction coefficient increased in the thermo-aerodynamic case compared to that of the gas rarefaction model, and thereafter it rose as the modulus of elasticity of the bearing continues to increased. This is due to the temperature profile across the microbearing leading to a higher molecular kinetic energy level. The friction comes from viscous shearing within the lubricant, and a higher shaft rotational speed means more viscous shear. In addition, the elastic deformation is a result of the aerodynamic pressure change in the bearing clearance. As pressure increases, the top portion of the bearing pad is likely to expand outwards in the pressurized region, which provides an increment in the internal rarefied gas flows.

### 4.2. Dynamic Stiffness and Damping Coefficients

Figure 11 indicates the variations of dynamic coefficients with dimensionless perturbation frequency *Ω* under different elastic deformation and temperature effects when *ε* = 0.7, *ω* = 8 × 10^4^ and *υ* = 0.3. It is clear that the direct stiffness coefficients *K_xx_* and *K_yy_* increased with an increase in the perturbation frequency. This is a consequence of enhancing the squeeze effect in the ultra-thin gas film, which is similar to the aerodynamic effect in the microbearing to some extent, and *K_yy_* is larger than *K_xx_* because the gas film mainly supports the weight of the high-speed rotor along the vertical direction. The direct damping coefficient *D_xx_* first increased steeply and reached its maximum at *Ω* = 1, then it decreased gradually in the range from *Ω* = 1 to 5, and *D_yy_* was smaller at the larger *Ω* for both cases. If thermal behavior of flows is regarded in the calculation, as a result of the strong dependence of gas lubricant viscosity on temperature, the stiffness coefficients for thermo-aerodynamic case were greater than that for gas rarefaction, while the effect of temperature decreased the direct damping coefficients of the microbearing during the change of perturbation frequency. For the thermo-elasto-aerodynamic solution, the stiffness coefficients further increased slightly with the growth of bearing pad flexibility, but it can cause an increase in the dynamic damping coefficients for a reduced elastic modulus. This difference is attributed to the fact that the surface elastic deformation and higher gas viscosity contribute to increase the rarefied Poiseuille flow in the clearance space.

Figure 12 shows the relationship between the dynamic coefficients and rotor eccentricity ratio with *Ω* = 4, *ω* = 8 × 10^4^ and *υ* = 0.3 for diverse lubrication conditions. As the eccentricity ratio increased, the direct stiffness coefficients *K_xx_* and *K_yy_* increased, and the direct damping coefficients *D_xx_* and *D_yy_* decreased. Additionally, *K_xx_* and *K_yy_* for microbearings with the non-uniform film temperature field were always bigger than those for the isothermal and constant viscosity gas lubricant films, whereas the opposite trend was displayed in the direct terms of damping coefficients. The reason might be that the heat transport by gas molecule collision, hence more energy, is transferred to the rarefied gas flow, which can, to some extent, compensate for the increase of the Knudsen number and the degree of gas rarefaction as a result of the increased temperature. It can also be seen that with the modulus of elasticity *E* decreasing, the direct stiffness coefficients increased, and direct damping coefficients decreased steadily in the thermo-elasto-aerodynamic solution. With lower *E,* the deformation effect became relatively more pronounced compared to the thermal effect, owing to the gas film thickness being the most modified.

The effect of rotor rotation speed *ω* on the stiffness and damping coefficients is shown in Figure 13 for each case. The results show that the direct stiffness coefficients increased, and the direct damping coefficients firstly increased rapidly and then decreased after reaching the maximum values with increasing *ω*. It is noteworthy that the difference in the continuum flow, rarefaction modified flow, thermo-aerodynamic and thermo-elasto-aerodynamic models started to diverge when the journal rotation speed exceeded 2 × 10^4^ rad/s. This is a consequence of stiffening the lubricant film because the aerodynamic effect of the rotation generates an ever-increasing perpendicular force component, which can prevent substantial energy dissipation. The gas film temperature in the lubricated region increased differently with varying flow conditions at high rotation speed. Decreasing values of elastic modulus *E* resulted in markedly increased values of stiffness coefficients *K_xx_* and *K_yy_,* while it decreased the direct damping coefficients *D_xx_* and *D_yy_*. The reason is attributed to the film thickness distribution, which has two converging diverging profiles in the thin elastic microbearing pad material, and this has less restriction on the pressure flow component in the sliding direction developed in the ultra-thin gas film.

## 5. Conclusions

To highlight the interactive effects of gas rarefaction, temperature, lubricant viscosity and the microbearing surface compliance in the compressible lubrication analyses for MEMS devices, the modified Reynolds equation, modified energy equation, elasticity equation and temperature–viscosity relationship are solved simultaneously. The modified Reynolds equation is solved using the relaxation iterative scheme and partial derivative method, and the finite element approach is utilized to compute the elastic distortions induced by the aerodynamic pressure. At the same time, the temperature distribution within the gas film is calculated by the finite difference formulation in the thermo-elasto-aerodynamic model. The following major conclusions can be drawn:(1)The peak temperature occurs at the vicinity of the location where the minimum gas film thickness appears. The temperature of the lubricant increases more significantly at higher eccentricity ratios than at higher speeds, and the elastic bearing pad is found to decrease the maximum gas film temperature.(2)As the eccentricity ratio and shaft rotation speed increase, both the load capacity and friction coefficient increase monotonically, while the attitude angle becomes smaller as the eccentricity ratio increases. The presence of thermal and elastic deformation effects increases the load capacity and friction coefficient because of the promoted gas viscosity and the enhanced aerodynamic phenomenon as well as the decreased rarefaction effect.(3)The direct stiffness coefficients increase and the direct damping coefficients begin to decrease for higher values of perturbation frequency, eccentricity ratio and rotor rotation speed. The direct stiffness coefficients increase with the increase of lubricant temperature; however, its effect on direct damping coefficients is reversed. The impact of elastic distortion of the bounding solids on dynamic coefficients in thermo-elasto-aerodynamic analysis is similar to the thermo-aerodynamic results with reduced elastic modulus. The thermo-elasto-aerodynamic behavior in the micro gas bearing is important to know to fundamentally understand the lubrication conditions in the bearing–journal pair.

## Figures and Tables

**Figure 1 micromachines-11-00955-f001:**
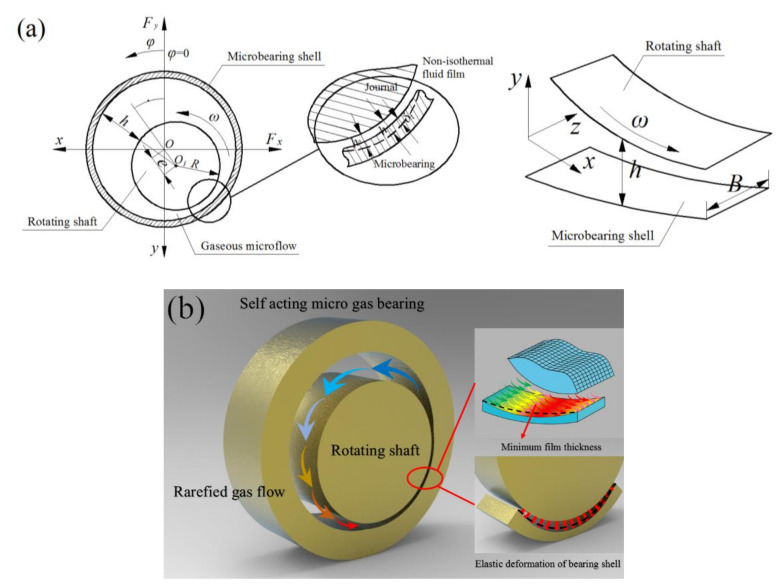
Schematic of the gas-lubricated microbearing: (**a**) schematic representation of a micro gas journal bearing and coordinates of the gas film; (**b**) three-dimensional graph.

**Figure 2 micromachines-11-00955-f002:**
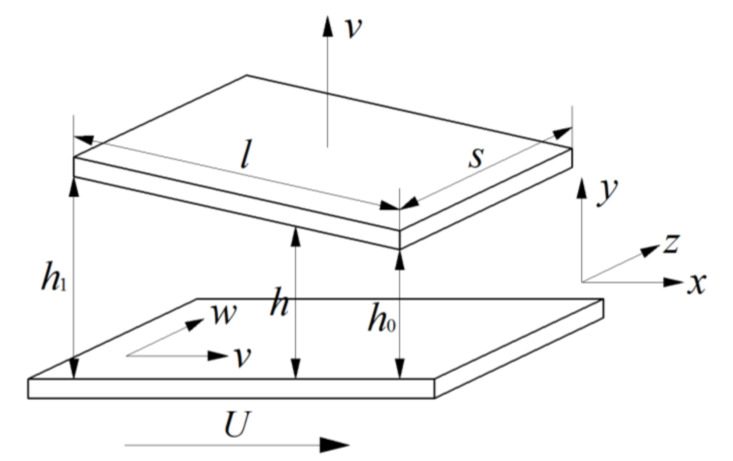
Coordinates and dimensions employed by Hwang et al. (1996).

**Figure 3 micromachines-11-00955-f003:**
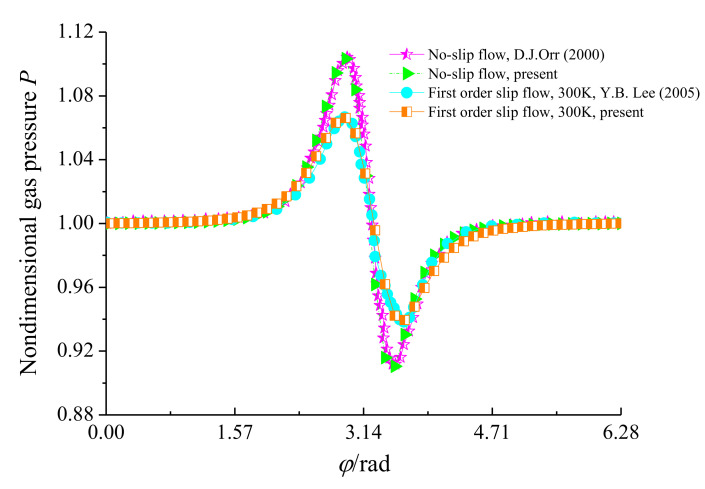
Comparison of Lee et al. and Orr’s simulation results to present predicted pressure distributions along circumference.

**Figure 4 micromachines-11-00955-f004:**
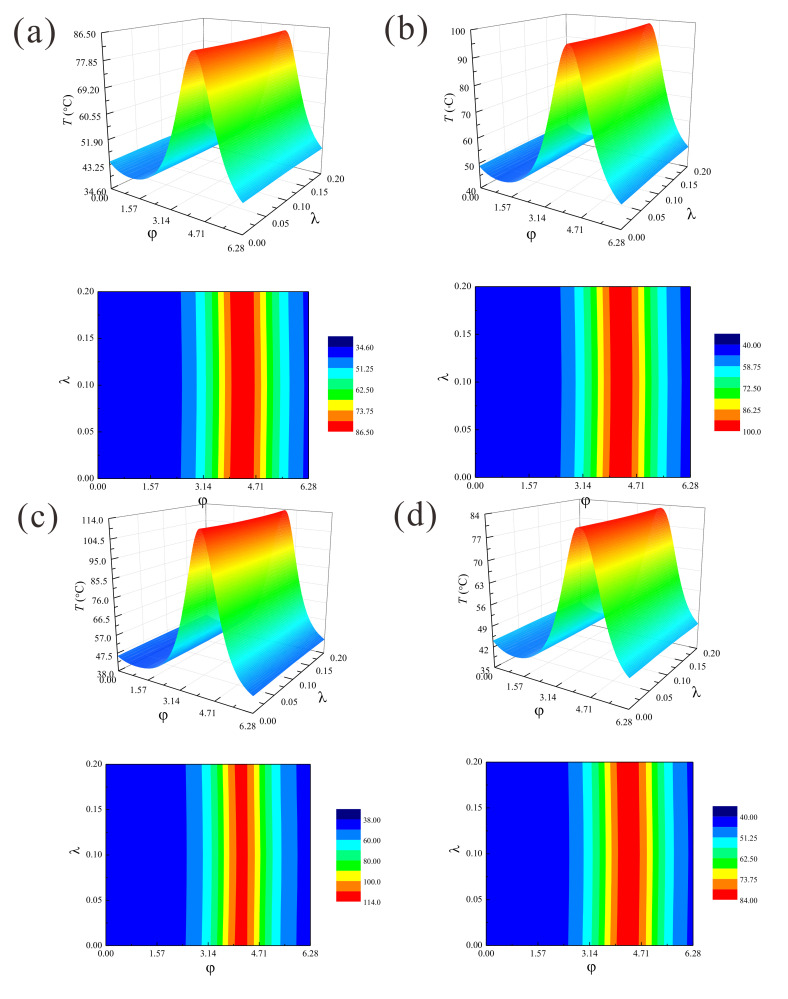
Temperature distributions and contours of temperature rise in the gas journal microbearing: (**a**) *ε* = 0.5, *ω* = 5 × 10^4^ rad/s, rigid; (**b**) *ε* = 0.5, *ω* = 6 × 10^4^ rad/s, rigid; (**c**) *ε* = 0.6, *ω* = 5 × 10^4^ rad/s, rigid; (**d**) *ε* = 0.5, *ω* = 5 × 10^4^ rad/s, elastic, *E* = 5GPa, *υ* = 0.3.

**Figure 5 micromachines-11-00955-f005:**
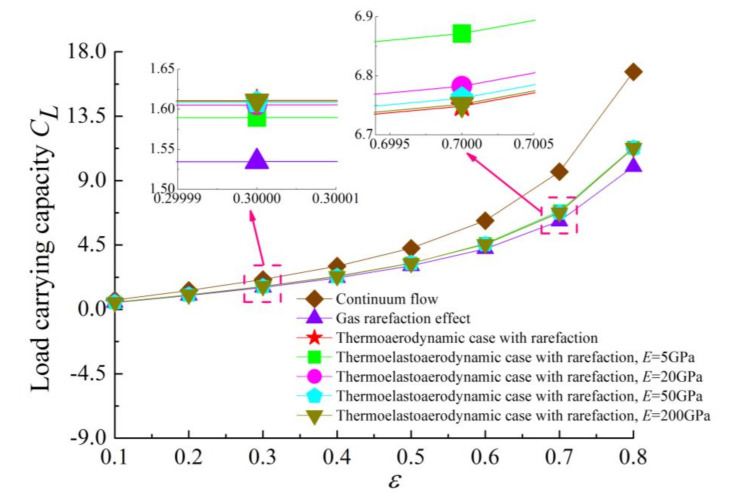
Variation of dimensionless load carrying capacity as a function of eccentricity ratio for different cases with *ω* = 7 × 10^4^ rad/s and *υ* = 0.3.

**Figure 6 micromachines-11-00955-f006:**
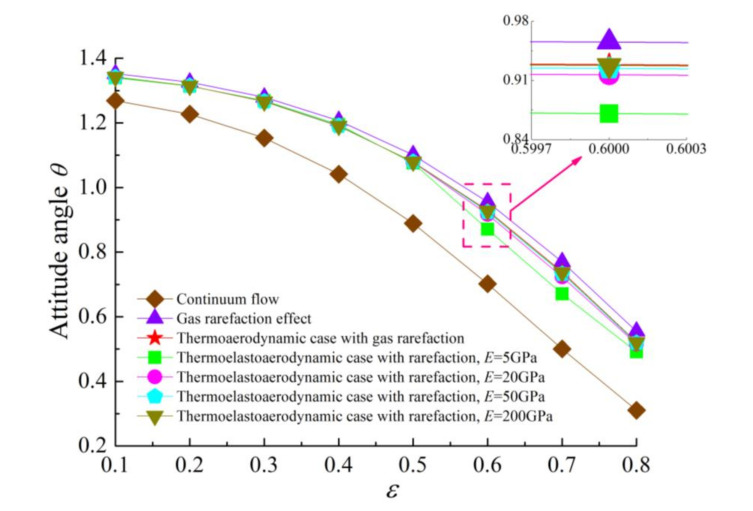
Attitude angle *θ* versus eccentricity ratio *ε* for different cases with *ω* = 7 × 10^4^ rad/s and *υ* = 0.3.

**Figure 7 micromachines-11-00955-f007:**
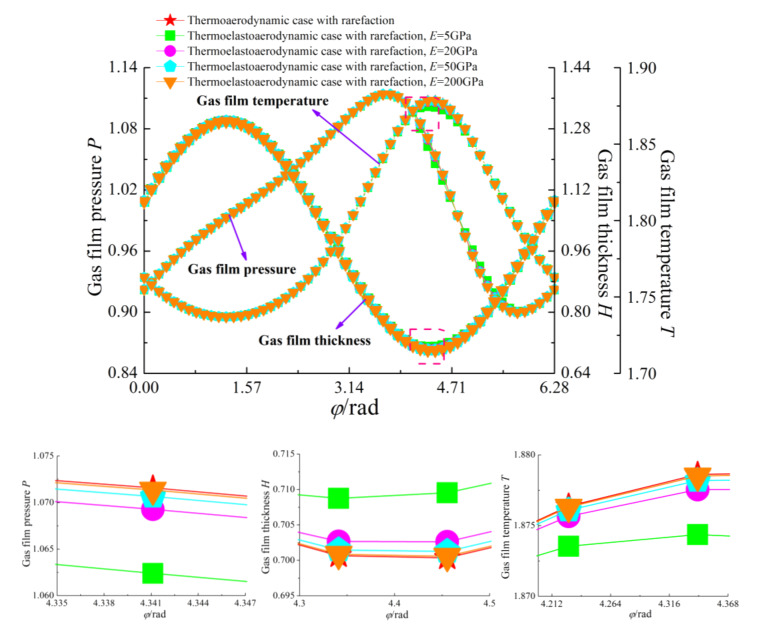
Magnified view of the dimensionless pressure, film thickness and temperature in terms of load carrying capacity for different cases with *ε* = 0.3, *ω* = 7 × 10^4^ rad/s and *υ* = 0.3.

**Figure 8 micromachines-11-00955-f008:**
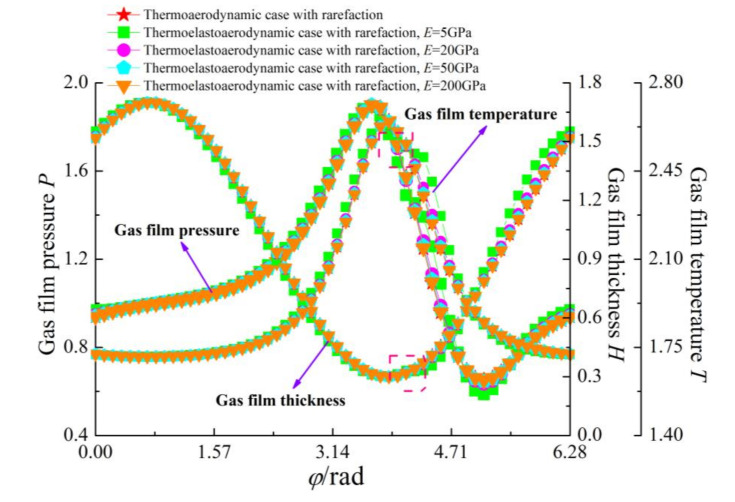
Magnified view of the dimensionless pressure, film thickness and temperature in terms of load carrying capacity for different cases with *ε* = 0.7, *ω* = 7 × 10^4^ rad/s and *υ* = 0.3.

**Figure 9 micromachines-11-00955-f009:**
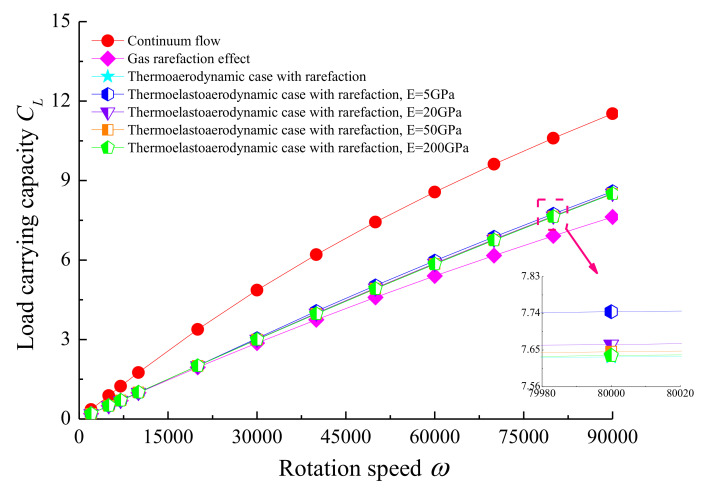
The relationship between load carrying capacity, *C_L_*, and shaft rotation speed, *ω*, for various cases with *ε* = 0.7, *υ* = 0.3.

**Figure 10 micromachines-11-00955-f010:**
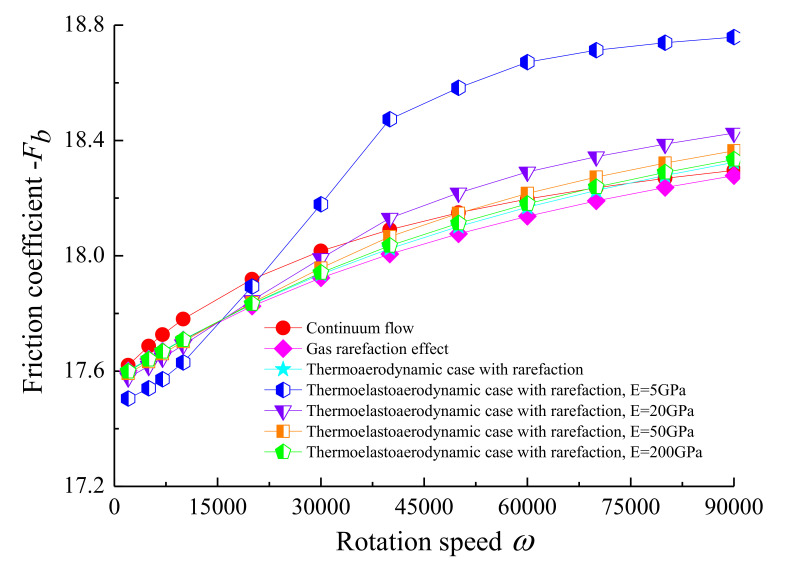
Friction coefficient against rotational speed of the journal with *ε* = 0.7, *υ* = 0.3 for different cases.

**Figure 11 micromachines-11-00955-f011:**
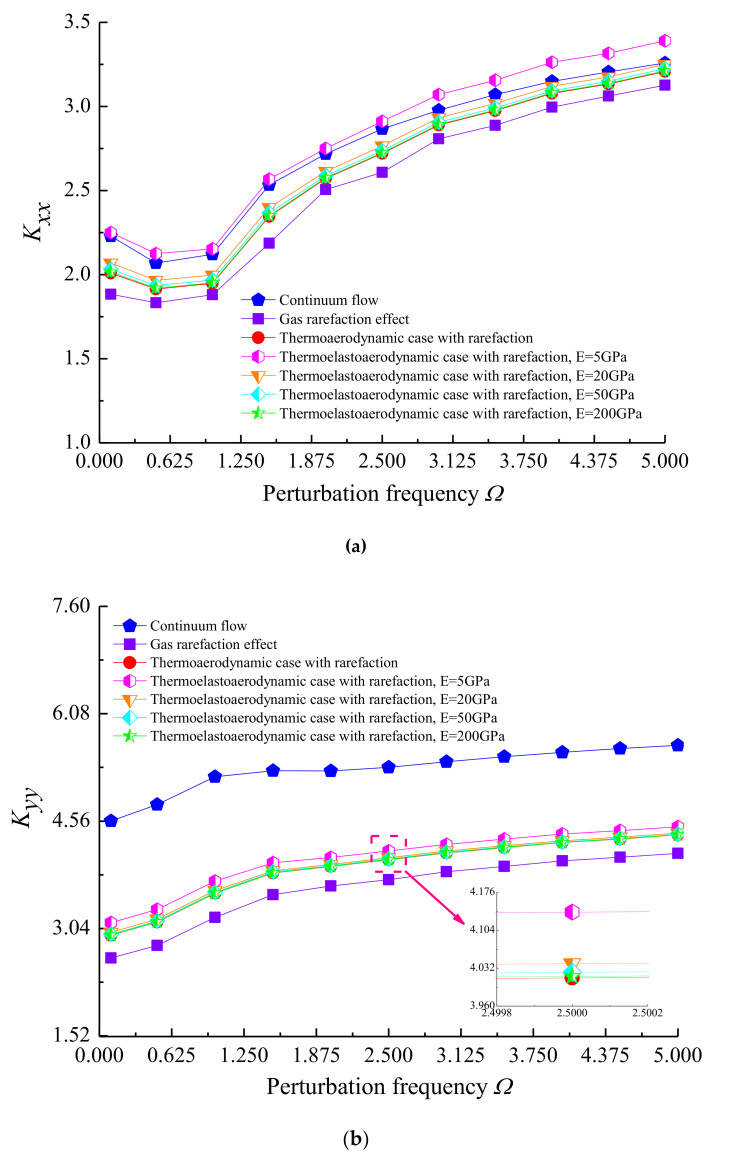
Effect of perturbation frequency on dynamic stiffness and damping coefficients for different cases with *ε* = 0.7, *ω* = 8 × 10^4^, *υ* = 0.3. (**a**) *K_xx_* vs. *Ω*; (**b**) *K_yy_* vs. *Ω*; (**c**) *D_xx_* vs. *Ω*; (**d**) *D_yy_* vs. *Ω*.

**Figure 12 micromachines-11-00955-f012:**
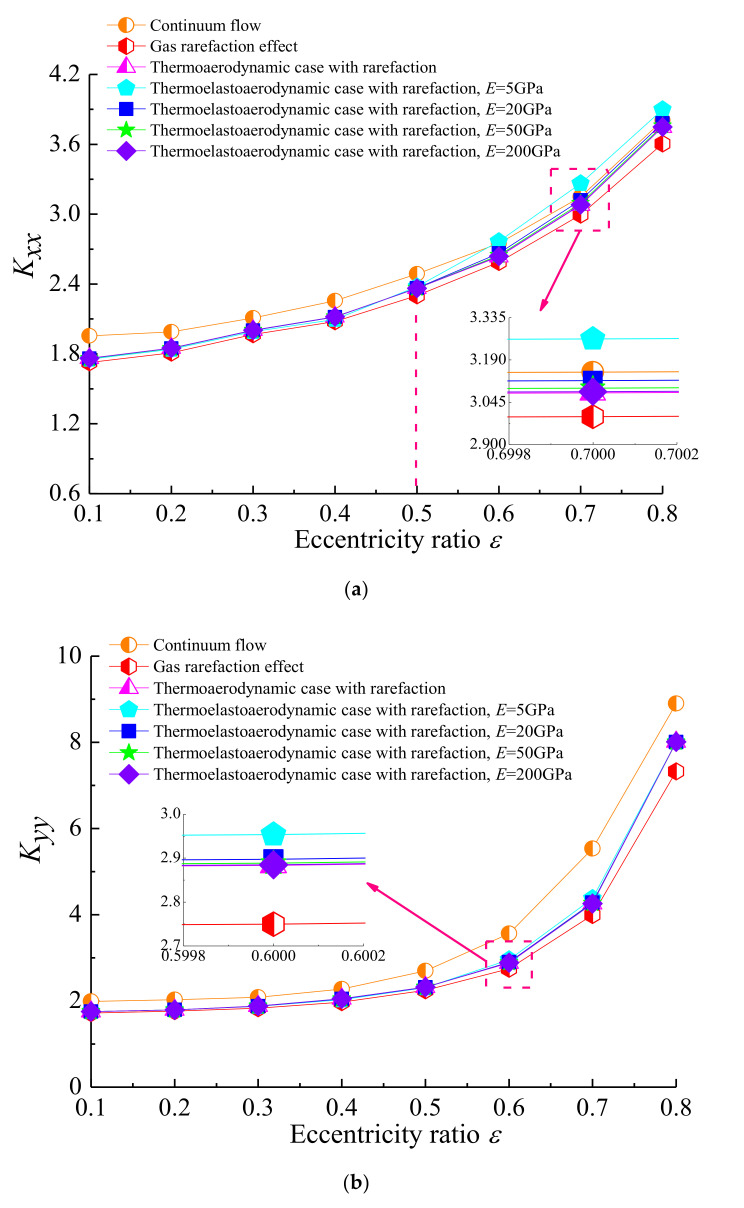
Effect of eccentricity ratio on dynamic stiffness and damping coefficients for different cases with *Ω* = 4, *ω* = 8 × 10^4^, *υ* = 0.3. (**a**) *K_xx_* vs. *ε*; (**b**) *K_yy_* vs. *ε*; (**c**) *D_xx_* vs. *ε*; (**d**) *D_yy_* vs. *ε*.

**Figure 13 micromachines-11-00955-f013:**
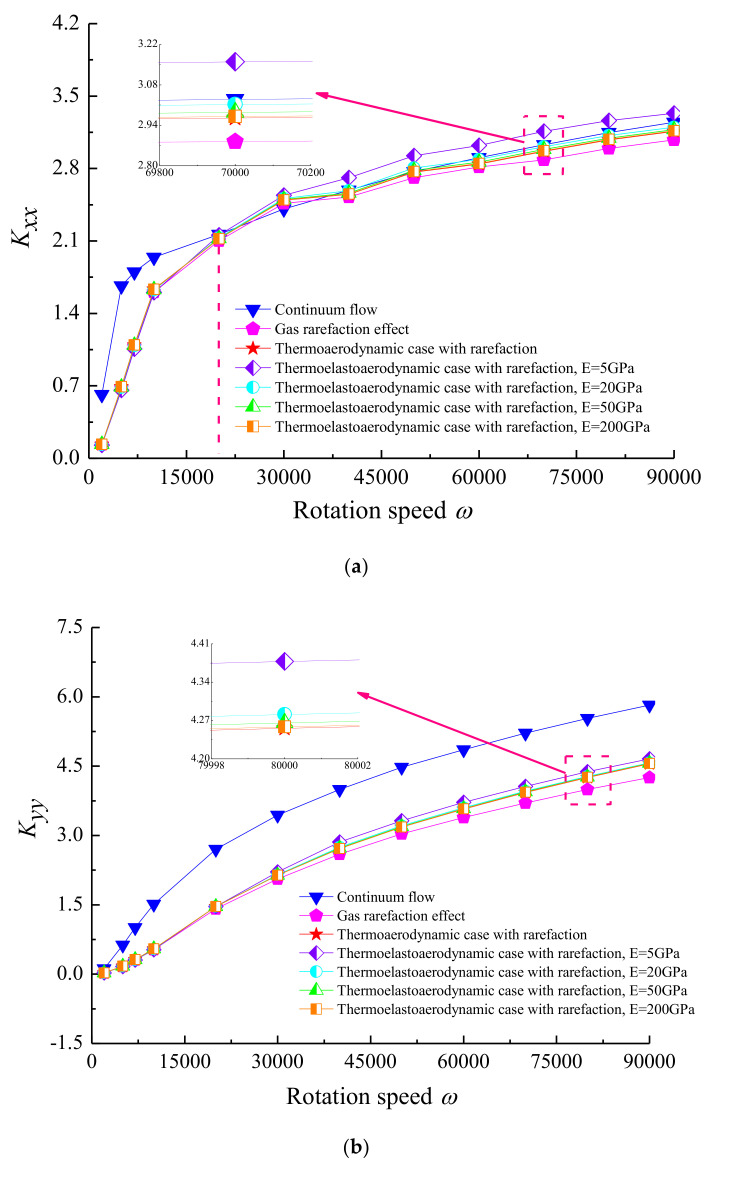
Effect of shaft rotational speed on dynamic stiffness and damping coefficients for different cases with *Ω* = 4, *ε* = 0.7, *υ* = 0.3. (**a**) *K_xx_* vs. *ω*; (**b**) *K_yy_* vs. *ω*; (**c**) *D_xx_* vs. *ω*; (**d**) *D_yy_* vs. *ω*.

**Table 1 micromachines-11-00955-t001:** Structure parameters of the microbearings.

Bearing Parameters	Values
Bearing radius *R* (mm)	1
Bearing length *B* (mm)	0.2
Ambient pressure *p_a_* (Pa)	1.01 × 10^5^
Poisson’s ratio *υ*	0.3
Clearance spacing of the gas film *c* (µm)	1
Ambient temperature *T*_0_ (K)	293.15

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
