# Peer review of "Thermo-Elasto-Hydrodynamic Characteristics Analysis of Journal Microbearing Lubricated with Rarefied Gas"

_micromachines, 2020, doi:10.3390/mi11110955_

Round 1

Reviewer 1 Report

The present paper investigates the thermal, elastic, and aerodynamic behaviors of a rarefied gas-lubricated journal microbearing. The authors have tested and discussed different parameters involved in this small-size lubrication such as the effects of eccentricity ratio, shaft speed, and bearing pad flexibility...

The results of this study are promising, well presented, and discussed. So, the paper is of high scientific interest for readers. For this reason, I support the publication of this nice paper in the Micromachines journal with minor revision.  In the following some recommendations:

  • As in the common literature related to the rarefied or vacuum gas flows, the symbol \lambda is generally reserved to the gas mean free path. So, please change the notation of the axial coordinate by \eta for example, or another.
  • Indicate the description of some symbols in the first use, like W and \mu_0 ...
  • the symbols q_x and q_z can be confused with heat flux vector components.
  • It is interesting to describe your numerical approach, mesh used, convergence criteria, time of computation, ... 
  • You should also conduct a mesh independence test. 
  • To more analyze the results, I think that is more convenient to use an appropriate scale for the same figures, like Fig1. 

Author Response

Dear Reviewer:

We would like to express our great appreciation to you for your help and valuable comments on our manuscript entitled “Thermoelastoaerodynamic Lubrication Analysis of Microbearings with Gas Rarefaction Effect” (Manuscript Number: Micromachines-943532). The comments are all instructive for revising and improving the present research work, as well as the important guiding significance to our future study. We have carefully addressed all the comments and have revised the manuscript which are marked in red color accordingly. Thank very much for your consideration and looking forward to hearing from you soon.

With kind regards,

Yours sincerely,

Yao Wu

Author Response

Dear Reviewer:

Thank you very much for your letter and for the important and instructive comments concerning our manuscript entitled “Thermoelastoaerodynamic Lubrication Analysis of Microbearings with Gas Rarefaction Effect” (Manuscript Number: Micromachines-943532). The comments are very valuable and helpful for revising and improving our paper. Accoring to the beneficial suggestions provided in your letter, We have studied comments carefully and have made correction which we hope meet with approval. Revised portion are marked in a different color (red) in the manuscript.

Best wishes,

Your sincerely,

Yao Wu

Reviewer 3 Report

The manuscript discusses the thermoelastoaerodynamic lubrication analysis of micro bearings with gas rarefaction effects. Given below are my comments to improvise upon the quality of the manuscript.

  1. The overall paper is well written and nicely structured. I would recommend adding a few statements on how the gas lubrication is advantageous and different from the conventional grease lubrication in the bearings. You may refer and cite this review paper

https://iopscience.iop.org/article/10.1088/1757-899X/149/1/012201/meta

  1. Please mention if the equations used in the work, are modeled by the authors or they are taken from some reference.
  2. You can consider changing the color scheme of fig 7 and 8. The data points are very bright and hard to read. Please make them similar to other figures for better visibility.
  3. Please consider rephrasing the title slightly, 'thermoelastoaerodynamic'    is a very long word to read altogether.  

Author Response

Dear Reviewer:

Thank you very much for your attention and comments concerning our manuscript entitled “Thermoelastoaerodynamic Lubrication Analysis of Microbearings with Gas Rarefaction Effect” (ID: Micromachines-943532). The comments are all enlightening and very helpful for improving the present research work, as well as the valuable guiding significance to our future researches. We have studied comments carefully and have made correction which we hope meet with approval. Revised portion are marked in red in the paper.

Best wishes,

Your sincerely,

Yao Wu

Round 2

Reviewer 1 Report

First, I would thank the authors for your careful revision and the response to my comments. The literature review is rather limited. For example, the authors should also refer to other numerical approaches that are commonly used for this kind of flows. The reviewer suggests that the following references: 

  • https://doi.org/10.1016/j.jcp.2012.11.023, R13 approach
  • https://doi.org/10.1155/2019/5084098, Moments method for thermally driven flows.      

After this suggestion, I support the publication of this interesting paper.

Author Response

Dear Reviewer:

I would like to thank you again for your help with our manuscript entitled “Thermoelastoaerodynamic Lubrication Analysis of Microbearings with Gas Rarefaction Effect” (ID: Micromachines-943532). According to your comments, I just resubmitted the second revision of manuscript. Thank very much for your consideration, and look forward to hearing from you soon.

Have a nice day!

Yours sincerely,

Yao Wu

Reviewer 2 Report

Authors should highlight or use different color for the text added in the revised version. In this revised paper, reviewer does not know where is the revised texts.

Besides, some figures and fomulars in the coverletter for reviewer were not added in the paper. Why?

In the Figure 4, authors should use same scale for the y-axis.

Author Response

Dear Reviewer:

Thanks for your help in our manuscript entitled “Thermoelastoaerodynamic Lubrication Analysis of Microbearings with Gas Rarefaction Effect” (Manuscript ID: Micromachines-943532). We have carefully addressed all the comments and revised the manuscript accordingly, and corresponding content are renewed in red color and in bold.

Thank very much for your consideration, and look forward to hearing from you soon.

With kind regards,

Yours sincerely,

Yao Wu

Reviewer 3 Report

Thank you very much for taking into consideration my comments. The overall quality of the manuscript is much improved. I could however spot a minor error in Ref 4. The correct reference is this:

Anand, G.; Saxena, P. A review on graphite and hybrid nano-materials as lubricant additives. IOP Conf. Ser. Mater. Sci. Eng. 2016149, 012201.

Kindly correct this in the final version.

Author Response

Dear Reviewer:

We would like to thank you again for the time and effort that you have put into reviewing the manuscript. The suggestions concerning our manuscript entitled “Thermo-elasto-aerodynamic Lubrication Analysis of Microbearings with Gas Rarefaction Effect” (ID: Micromachines-943532) are of the great help to our research work. We have studied comments carefully and have made correction which we hope meet with approval. Looking forward to hearing from you.

Wish you all the best!

Sincerely yours,

Yao Wu
